# Medication Adherence in Hypertensive Individuals in Panama 2019: A National Cross-Sectional Study

**DOI:** 10.3390/healthcare10112244

**Published:** 2022-11-09

**Authors:** Carlos Guerra, Eric Conte, Angela Isabel Del Rio, Jorge Motta, Ilais Moreno Velásquez, Hedley Knewjen Quintana

**Affiliations:** 1Department of Research and Health Technology Assessment, Gorgas Memorial Institute for Health Studies, Panama City 0816-02593, Panama; 2Ministry of Health, Panama City 0843-03441, Panama

**Keywords:** Panama, medication adherence, hypertension, cross-sectional study design

## Abstract

Poor medication adherence is a public health concern leading to a large burden of cardiovascular disease among persons with hypertension. Using data from 3281 persons with diagnosed hypertension (N = 622,581) from the Panamanian National Health Survey (ENSPA) collected in 2019, we assessed the national prevalence of low-moderate medication adherence in hypertensive individuals using the 4-scale Morisky Medication Adherence Scale (4-MMAS) and identifying gender-specific associated factors. Multivariate logistic regression models were used to estimate the association between possible risk factors and low-moderate medication adherence with odds ratios (OR) and 95% confidence intervals (95% CI) stratified by gender. The national prevalence of low-moderate medication adherence was 78.2% (95% CI: 74.7–81.0%); in men it was 74.4% (95% CI: 67.5–80.3%) and in women it was 81.4% (78.4–84.0%). In women, low-moderate medication adherence was associated with living in indigenous area (OR: 5.15; 95% CI: 1.40–18.98), educational level (OR no formal education: 0.77, 95% CI 0.28–2.14; OR for primary education: 0.76, 95% CI 0.38–1.56; OR for secondary education: 0.90, 95% CI 0.48–1.70; Higher education as reference), increased BMI (normal as reference, OR for overweight: 1.35, 95% CI: 0.73–2.50, OR for obesity: 1.65, 95% CI: 0.90–3.03) and medical diagnosis of anxiety/depression (OR: 4.89, 95% CI: 1.36–17.49). However, in men, it was associated with having secondary education (OR: 2.94; 95% CI: 1.03–8.36), currently smoking (OR: 16.74, 95% CI: 1.83–152.70), taking antihypertensive medication with denial of hypertension diagnosis (OR: 4.35, 95% CI: 1.11–17.11) and having less than three annual check-ups (OR for no health check-ups: 2.97, 95% CI: 0.63–13.88; OR for 1–2 check-ups: 1.61, 95% CI: 0.78–3.32: three or more health check-ups: reference). Time since diagnosis was inversely associated with low-moderate adherence. This study assesses for the first time the national prevalence of low-moderate medication adherence among hypertensive individuals in Panama. Low-moderate medication adherence is an important public health issue that should be addressed to achieve blood pressure control in patients diagnosed with hypertension, taking into account gender-specific factors.

## 1. Introduction

Hypertension is defined as having a systolic blood pressure (SBP) ≥140 mmHg and/or diastolic (DBP) ≥90 mmHg [1,2]. Hypertension is a condition affecting approximately 1.38 billion people worldwide [3], and it is a serious global public health problem, being a major risk factor for kidney and cardiovascular disease (CVD) which causes 8.5 million deaths worldwide annually, most of them occurring in low-income and middle-income countries (LMICs) [4,5]. The prevalence of hypertension is expected to rise globally [3,5] due to increasing prevalence of other cardiovascular risk factors such as obesity, high-sodium diet, excessive alcohol consumption, and physical inactivity [4].

Hypertension is one of the most important modifiable risk factors for developing premature CVD [6]. Patients with hypertension often lack symptoms, making its screening a need for its detection [7]. Once hypertension has been detected, its treatment and control are one of the most effective public health interventions to prevent CVD morbidity and mortality [6]. To achieve hypertension control requires adherence to medication, defined as the actions of the patient for taking it as prescribed by their health providers and agreed upon in the treatment plan [8]. Global prevalence of antihypertensive medication adherence has been estimated to be between 30% and 50% annually [6]. Nevertheless, according to various studies, the proportion of medication adherence varies across different countries, for example, from 85% in Australia, up to 38.8% in the USA [9]. Poor adherence accounts for a large public health burden: with estimated costs between $100 and $290 billion yearly [10]. According to the World Health Organization (WHO), a variety of reasons for the lack of adherence have been described, among them: social and economic factors (age, ethnicity, sex, socioeconomic and educational status), patient-related factors (health beliefs, health literacy, lack of knowledge, forgetfulness and fear of dependence), cost of treatment, adverse effects, comorbid conditions (alcohol and other drug abuse, depression, coexisting chronic medical conditions), and finally, health care system factors (patient-provider relationship, absent or limited care coordination and integration) [11].

Several methods have been developed to measure medication adherence. These methods can be grouped into three categories: subjective (self-report), direct (serum or urine drug level), and indirect (pharmacy database records, pharmacy refill rates or pill counts) [12]. Each one has its advantages and disadvantages; the subjective and indirect methods are the more frequently used in adherence-related series. Self-reported measures are simpler and inexpensive methods that may give more in-depth insights about the situational and behavioral factors influencing adherence [8].

In 2012, the prevalence of hypertension in two of the most populated provinces in Panama was 29.6% [13]. Later in 2019, the prevalence of hypertension according to the National Health Survey of Panama (ENSPA in Spanish Language) was 45% [14]. The medication adherence was less than 40% in hypertensive out-patients who attended four primary care health centers (city of Panama, Chiriquí, San Miguelito, Veraguas) using a 4-item Morisky Medication Adherence Scale (MMAS-4) in 2016 [15]. Nevertheless, to date, there is no recent nationwide study that has assessed adherence to medication in hypertensive individuals in the whole country. Understanding the prevalence of medication non-adherence and its associated factors is crucial for determining intervention strategies.

This study aims to report the national prevalence of medication adherence in hypertensive individuals and to identify associated factors using data from the National Health Survey for Panama (Encuesta Nacional de Salud de Panamá -ENSPA-), a cross-sectional study performed in 2019.

## 2. Materials and Methods

### 2.1. Study Materials

The ENSPA is a national population-based cross-sectional study that assessed health determinants, environmental, nutritional, anthropometric factors, as well as health service access in Panama during 2019. Eligible subjects were all individuals living at least 6 months in their households. The sampling frame was designed to ensure the representativeness of the population divided into two groups: 0–14 years old and 15 years old and older with a complex sampling design (a randomized tri-phased stratified by conglomerates). The sample plan applied for the ENSPA allowed the representativeness of the results to the whole country population up to its second-level administrative division (district) and up to its third-level administrative division (corregimiento) in the Panama and San Miguelito districts. Based on census estimates, households were randomly selected to achieve representativeness of national, provincial, indigenous, rural, and urban areas. Per household, up to two potential participants, one 15 years or older and another less than 15 years old (if available) were randomly selected. A total of 17,997 participants aged 15 years or more were selected, 96.1% of them being 18 years old or older. ENSPA was conducted by the Gorgas Memorial Institute of Health Research, the Ministry of Health of Panama, and the INEC (Spanish language for “National Institute of Statistics and Census”). The sampling design was calculated using population projection for 2019 using data from the latest two National Censuses (2000 and 2010). A detailed description of ENSPA is published in its website in the Spanish language [14] and in previous studies [16,17].

For this study, we included participants aged 18 years or more who declared a self-reported medical diagnosis of hypertension and/or who stated that they had been taking antihypertensive medication in the previous two weeks regardless of self-reported diagnosis (because there is no reason for taking such medications unless they have a medical diagnosis of hypertension).

### 2.2. Data Collection

Participants answered a questionnaire in the Spanish language through in-person interviews, which collected information on demographics, socioeconomics, medical diagnoses, family history and lifestyle factors, including the outcome of this study. After each interview ended, other trained personnel took blood pressure, height, weight, and abdominal circumference measurements.

All anthropometric measurements were described previously [14,16,17]. In brief, the participant’s weight was assessed using a portable digital scale and their height was assessed using a portable stadiometer. Participants were classified according to WHO (World Health Organization) body mass index (BMI) categories: normal weight (between 18.5 kg/m^2^ and 24.9 kg/m^2^), overweight (between 25 and 29.9 kg/m^2^), and obesity (30 kg/m^2^ or more).

### 2.3. Outcome Assessment

Adherence to therapy was assessed through the Spanish version of the semi-structured questionnaire MMAS-4 [18]. The MMAS-4 questionnaire consists of a set of four questions with a yes or no answer. The following questions were queried: (1) Do you ever forget to take your medicine? (2) Are you careless at times about taking your medicine? (3) When you feel better, do you sometimes stop taking your medicines? (4) Sometimes, if you feel worse when you take the medicine, do you stop taking it? One point was awarded for every yes response and zero for every no. The total MMAS-4 score ranged from 0 to 4: In this study, the MMAS-4 score was dichotomized into high adherence (score 0) and low-moderate adherence (1 points or more) as performed in previous studies where it was used [19,20,21,22,23].

### 2.4. Exposure Variables

Three exposure variable categories for medication adherence were identified in the ENSPA questionnaire: sociodemographic, condition-related and patient-related [11].

#### 2.4.1. Sociodemographic Variables

The exposures assessed were age (years), region (urban, rural and indigenous), and highest attained education level (no formal education, primary, secondary, and higher).

#### 2.4.2. Condition-Related Variables

Smoking tobacco use was assessed according to whether the individual was a current smoker or not (smoked in the previous 30 days). Self-reported medical history of NCDs with similar medications to hypertension if the participant reported that he/she was diagnosed with the following conditions: chronic kidney disease (CKD), cerebrovascular disease (CVD), and acute myocardial infarct (AMI). Anxiety or/and depression was defined in an alike fashion. We excluded all the participants who reported any comorbidities, except self-reported medical history of CKD, CVD, AMI, anxiety and/or depression. CKD, CVD, AMI have similar treatment to hypertension. Anxiety and depression have been assessed as risk factors for poor adherence.

#### 2.4.3. Patient-Related Variables

Time since hypertension diagnosis was categorized as: “Antihypertensive medication intake at least two weeks before the interview but denying hypertension self-reported medical history”, “Less than a year before the interview”, “1–5 years before the interview” and “≥6 years before the interview”.

The participants were queried how many health check-ups they had undergone in the last year. The possible answers for this question were: “no check-ups”, “one or two checkups” and “three or more checkups” in the last year by a physician.

### 2.5. Statistical Analysis

We report the number of persons with the outcome (*n*) and the population they represent (N). Categorical variables are presented as percentages with the corresponding 95% confidence intervals (95% CIs). Age was presented as median and interquartile range (IQR). For both continuous and categorical variables, their description considered the complex sampling design. Participants with missing data were excluded from the analysis.

In the literature, the following variables were associated with low-moderate adherence: age, area, education, current activity, BMI, CVD comorbidities, anxiety/depression, and the time from hypertension diagnosis [11]. To examine the association of each exposure variable with a low-moderate MMS-4 score, its respective prevalence odds ratios (OR) with their 95% CI were calculated by applying unconditional logistic regression models. Multivariable logistic regression models were performed using variables known to be independent predictors of low-moderate medication adherence in the participants with hypertension and stratified by sex, taking into account sampling weights.

Data were analyzed with the “survey” package version 4.1.1 and R version 4.2.1.

## 3. Results

### 3.1. Participation Flowchart

The flowchart summarizing the inclusion criteria and exclusion criteria are depicted in Figure 1. In this study, there were 3281 persons with hypertension (N = 622,581); amongst them 36.4% lack information on medication adherence. Hypertensive participants lacking information on medication adherence are described in Appendix A. Those who had missing data on MMAS-4 were more likely to be younger, current smokers, and to have fewer yearly check-ups and a higher monthly family income, compared to those who replied to the questionnaire.

### 3.2. Baseline Characteristics

The baseline characteristics of 3281 hypertensive participants are shown in Table 1. 52.8% of the participants were men and 48.2% were women. Their median age was 61 years (IQR 49–72). Overall, the predominant ethnicity was mestizo (women: 52.5% and men: 46.3%), and most lived in an urban region 73.2% (73.5% men vs. women 72.8%). According to the education levels, 4.4% reported having no formal education (women: 4.1% vs. men: 4.8%), 43.3% reported having secondary education (women: 43.2% vs. men: 43.5%), and 19.3% reported having higher education; men reported more often than women (22.1% vs. 16.9%). The prevalence of current smoker was 4.2% overall (women: 2.4% vs. men: 6.2%). Obesity was found in 43.6% of participants [women: 50.7% (95% CI: 47.3–54%); men: 35.4% (95% CI: 29.8–41.5)]. Excessive alcohol drinking was 7.1% overall [women: 8.3% (6.2–10.69); vs. men: 5.3% (3.3–8.3)]. The prevalence of self-reported medical history of CKD was 0.7%; for CVD it was 2.1%, and AMI it was 1.6%. In contrast, women had more anxiety and/or depression compared o men (women: 3.9%; men: 2.1%). The percentage of participants who denied self-reported medical diagnosis but were taking hypertension medication was 35.0%.

The national and regional items of the MMAS-4 score and the medication adherence are shown in Table 2. The national low-moderate adherence rate was 78.2% (IC 95%: 74.7–81.0%) without statistically significance differences per area, but with a tendency for this proportion to be lower in the indigenous area 87.4% (65.5–96.2%) than other areas. The item with the most common affirmative responses nationwide, in the urban and the rural areas, was “*Are you careless at time about taking your medicine?”* (national proportion: 51.1%, 95% CI: 47.3–54.9%), and the item “*When you feel better, do you sometimes stop taking your medicines?*” had the least frequent affirmative reply (national proportion: 24.0%, 95% CI: 20.9–26.8%). In contrast, the latter was the one with most common affirmative replies in the indigenous area (69.9%, 95% CI: 52.2–83.2%), and the item “*Sometimes, if you feel worse when you take the medicine, do you stop taking it?*” was the item with the least common affirmative responses (27.8%, 95% CI: 16.1–43.5%).

The prevalence of low-moderate medication adherence by sex and selected variables are presented in Table 3. Women have a higher prevalence of low-moderate medication adherence than men. The prevalence of low-moderate medication-adherence prevalence according to different variables under study is always above 50% in each group, ranging between 52% (95 IC%: 28.0–75.2%) in males with CKD, CVD, and/or AMI, up to 98.5% in men who currently smoke. In women, those with anxiety/depression had the highest prevalence (89.7%, 95 IC%: 73.7–96.4%) and those with higher education had the lowest (67.2%, 95% CI: 48.6–81.6%).

### 3.3. Predictors of Low-Moderate Medication Adherence

The crude and adjusted predictors of low-moderate adherence stratified by sex are shown in Table 4.

In the multivariate model for women, it was observed that living in an indigenous area (OR: 5.15; 95% CI: 1.40–18.98), increased BMI, and having self-reported medical history of anxiety and/or depression were associated with an increased occurrence of low-moderate medication adherence. However, in men, having secondary education (OR: 2.94, 95% CI; 1.03–8.36), current smoking (OR: 16.74, 95% CI: 1.83–152.70), recent self-reported medical diagnosis of hypertension (OR for no hypertension self-report, but taking hypertension medication: 4.35, 95% CI: 1.11–17.11; OR for less than a year before the interview: 1.14, 95% CI: 0.37–3.52; OR for 1–5 years before the interview: 1.14, 95% CI: 0.37–3.52), and fewer than three health checkups yearly were associated with an increased occurrence of low-moderate medication adherence.

## 4. Discussion

The aim of this study was to assess the prevalence of low-moderate medication adherence using the MMAS-4 among hypertensive individuals in the Republic of Panama, and gender-specific factors.

The rates of adherence vary between countries due to different types of population, study designs, and scales used to measure medication adherence [9,21,22,23,24,25,26,27,28,29,30,31,32]. Despite the wide variability of methods assessing medication adherence around the world, the low-moderate adherence prevalence found in our study was similar to that of persons diagnosed with hypertension in Latin American cities [23,24,25,26,27,32], as well as other populations from other places in the world such as Africa [30,31] and Asia [28,33]. A 2015 Spanish study using MMAS-4 showed a better adherence than our results [22].

There is a large body of literature stating that sex and gender render inequities in health [33,34]. However, few studies on medication adherence address such inequities. Our results show that risk factors for low-moderate adherence were different for both genders. For instance, in men, low-moderate adherence was associated with a sociodemographic factor: secondary education; a condition-related factor: currently smoking; and two patient-related factors: taking hypertensive medication while denying hypertension diagnosis, and having fewer than three yearly check-ups. Secondary education in men as a risk factor for low-moderate adherence might indicate that the understanding of hypertension as a life-threatening condition, as communicated by the health care team, seems to be low, but in women such an understanding decreases as the educational level increases. Our results contrast with other studies that have shown that educational level was not associated with medication adherence [28,35].

Our results in men on the effect of current smoking are similar to those reported in Ethiopia (in both men and women) regarding treatment compliance [36]. However, we did not observe such an effect in women.

Increased BMI is a well-known factor for increased mortality [37]. Despite overweight and obesity having metabolic effects, as well as causing changes in the pharmacodynamics of medications, our results also suggest they are associated with low-moderate adherence only in women. We might explain this gender-differential finding as a consequence of women having a poor body self-perception as BMI increases [38], and in turn, such self-perception might be associated with poor medication adherence [39]. BMI extracted from physical examination in women seems to be valuable predictor regarding medication use.

Among the patient-related factors, anxiety and depression were associated with increased low-moderate adherence in women, but not in men. Our results also show that taking antihypertensive medication while denying a hypertension diagnosis increases low-moderate adherence in men, but not in women. The national rules state that patients with hypertension must have a check-up twice a year [40], but our results indicate that men were 61% less likely to adhere to medication regimes compared to those attending three times per year. Therefore, our results suggest that psychiatric history in women and at least three check-ups are needed to increase medication adherence in patients taking antihypertensive medications.

Time from hypertension diagnosis was related to improved medication adherence. This variable was the only one that was present in both sexes. Unfortunately, it means that, after adjustment for other variables, it takes time to be diagnosed, so the patient gets confident enough to follow his/her attending physician’s recommendations. However, until such time passes, hypertension leads to potential damage, and the health authorities must deal with such a delay in treating the condition.

Among the limitations of this study, firstly, we used an indirect method for evaluation of the factors that predict adherence through self-questionnaires; since this study is a population-based survey, using assessments via direct methods or more specific in-depth questionnaires would make the survey impossible to perform. Nonetheless, the ENSPA questionnaire was applied by trained staff in a standardized fashion. Furthermore, our results on adherence are quite similar to results from other Latin American cities where similar methods were applied [21,23,24,25,27]. Secondly, ENSPA did not have therapy-related questions, enquiring for example into a specific antihypertensive treatment scheme, so we analyzed a sample of those participants with medical self-diagnosis and those who took antihypertensives without other comorbidities (except comorbidities like cerebrovascular disease, AMI, chronic kidney disease, anxiety and/or depression). Thirdly, potential biases have been introduced by the high number of missing values in our study. Finally, this study has a cross-sectional design and thus, causality should not be inferred.

The strength of this study was the large study population investigated using a survey that represents the national population of persons with hypertension diagnosis, as well as the sampling methodology that allowed us to provide robust information.

## 5. Conclusions

This is the first national study in the Republic of Panama assessing low-moderate medication-adherence prevalence in participants with hypertension. However, our results indicate that low-moderate medication adherence is an important public health issue that should be addressed to achieve blood pressure control in patients diagnosed with hypertension and considering gender-specific factors.

## Figures and Tables

**Figure 1 healthcare-10-02244-f001:**
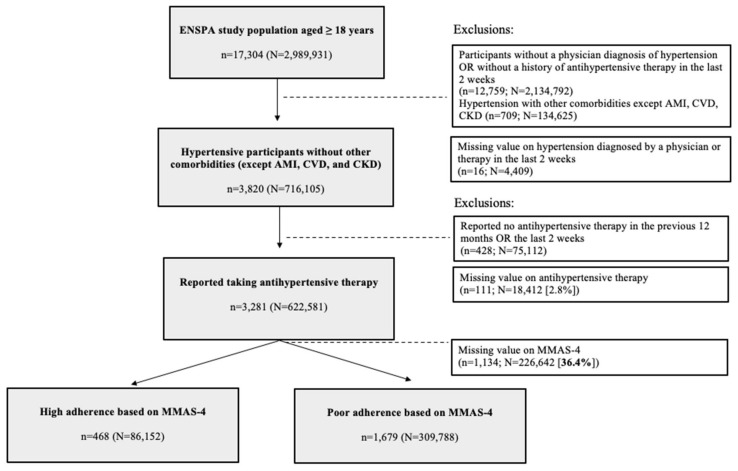
Flowchart of selected participants. *n* = study population. N = weighted study population. AMI = acute myocardial infarction. CVD = cerebrovascular disease. CKD = chronic kidney disease. MMAS-4 = Morisky medication adherence score-4 items. High Adherence defined as an MMAS-4 = 0, Low-moderate Adherence defined as an MMAS-4 > 0.

**Table 1 healthcare-10-02244-t001:** Baseline characteristics among treated hypertensive participants from the ENSPA study stratified by sex.

	National N = 622,581 % (95% CI)	Women N = 334,107 % (95% CI)	Men N = 288,474 % (95% CI)
**Sociodemographics**			
Age in years median (IQR)	61 (49–72)	59 (47–70)	63 (52–73)
Ethnicity			
Caucasian	23.5 (20.8–26.6)	23.0 (20.4–25.8)	24.2 (19.2–30.0)
Afro-Panamanian	18.4 (16.1–21.1)	16.6 (14.3–19.1)	20.6 (16.4–25.6)
Mestizo	49.6 (46.6–52.7)	52.5 (49.4–55.7)	46.3 (40.9–51.8)
Indigenous	4.7 (3.7–5.9)	4.7 (3.6–6.2)	4.7 (3.1–7.0)
Asians and others	3.7 (2.7–5.0)	3.2 (2.2–4.6)	4.2 (2.6–6.9)
Regions			
Urban	73.2 (70.8–75.4)	73.5 (71.1–75.8)	72.8 (68.4–76.7)
Rural	25.5 (23.3–27.8)	25.6 (23.4–28.0)	25.3 (21.5–29.6)
Indigenous	1.4 (1.0–1.8)	0.9 (0.6–1.3)	1.9 (1.3–2.9)
Education			
Higher education	19.3 (16.6–22.2)	16.9 (14.6–19.4)	22.1 (17.2–27.9)
Secondary education	43.3 (40.3–46.4)	43.2 (40.1–46.4)	43.5 (38.0–49.1)
Primary education	33.0 (30.3–35.7)	34.8 (32.9–38.8)	29.7 (25.2–34.5)
No education	4.4 (3.6–5.5)	4.1 (3.3–5.2)	4.8 (3.3–6.9)
**Condition-related**			
Current smoker	4.2 (3.0–5.8)	2.4 (1.5–3.7)	6.2 (4.0–9.6)
BMI categories			
Normal (18.5–24.9 kg/m^2^)	20.3 (17.7–23.2)	25.6 (13.2–18.0)	25.4 (20.7–30.7)
Overweight (25–29.9 kg/m^2^)	35.4 (32.4–38.6)	32.2 (29.2–35.4)	38.1 (32.7–43.7)
Obese (≥30.0 kg/m^2^)	44.2 (41.0–47.5)	50.7 (47.3–54.0)	35.4 (29.8–41.5)
Alcohol consumption			
Non-drinker	88.0 (85.7–90.0)	93.0 (91.1–94.6)	82.2 (77.7–86.0)
Moderate drinker	4.9 (3.6–6.5)	3.0 (1.9–4.5)	7.1 (4.8–10.3)
Excessive drinker	7.1 (5.6–9.0)	8.3 (6.2–10.9)	5.3 (3.3–8.3)
**Patient-related**			
**Self-reported comorbidities**			
CKD, CVD and/or AMI	4.2 (3.0–5.9)	2.6 (1.7–3.8)	6.2 (3.9–9.6)
Anxiety and/or depression	3.0 (2.1–4.4)	3.9 (2.7–5.4)	2.1 (0.8–5.1)
**Time from hypertension diagnosis**			
Antihypertensive medication intake two weeks before the interview	35.0 (32.0–38.1)	33.6 (30.7–36.7)	36.7 (31.3–42.4)
Less than a year before the interview	9.3 (7.7–11.1)	9.0 (7.4–10.8)	9.6 (7.0–13.2)
1–5 years before the interview	19.5 (17.2–21.9)	19.5 (17.2–22.1)	19.4 (15.5–24.0)
≥6 years before the interview	36.2 (33.4–39.1)	37.9 (34.9–41.0)	34.3 (29.4–39.5)
**Health checkups frequency**			
None	14.7 (12.6–17.2)	11.5 (9.2–14.4)	20.5 (16.4–25.3)
1–2 yearly	57.6 (54.5–60.6)	55.9 (52.1–59.6)	60.4 (55.1–65.4)
≥3 yearly	27.7 (25.1–30.4)	29.1 (26.4–32.0)	26.1 (21.6–31.1)

N: weighted study population; % = percentage. CI = confidence intervals. BMI = body mass index; CKD: chronic kidney disease; CVD = cerebrovascular disease. AMI = acute myocardial infarction.

**Table 2 healthcare-10-02244-t002:** Adherence to medication among treated hypertensive individuals using the 4-item Morisky Medication Adherence Scale from the ENSPA study, stratified by region.

	National N = 622,581 % (95% CI)	Urban N = 455,434 % (95% CI)	Rural N = 158,649 % (95% CI)	Indigenous N = 8498 % (95% CI)
**The 4-items MMAS**				
Replied **yes** to query *“Do you ever forget to take your medicine?”*	33.6 (30.1–37.4)	32.9 (28.3–37.7)	35.3 (30.6–40.4)	43.3 (25.5–63.1)
Replied **yes** to query *“Are you careless at time about taking your medicine?”*	51.1 (47.3–54.9)	52.3 (47.4–57.2)	48.8 (43.7–53.9)	31.5 (17.1–50.8)
Replied **yes** to query “*When you feel better, do you sometimes stop taking your medicines?*”	24.0 (20.9–26.8)	21.9 (18.4–25.8)	25.8 (21.5–30.6)	69.9 (52.2–83.2)
Replied **yes** to query “*Sometimes, if you feel worse when you take the medicine, do you stop taking it?*”	41.6 (38.0–45.2)	40.4 (35.9–45.0)	45.8 (40.8–50.9)	27.8 (16.1–43.5)
**Low-moderate adherence** **MMAS:** at least a **“yes”** reply to any of the four queries	78.2 (74.7–81.0)	77.8 (73.2–81.8)	79.0 (74.6–82.8)	87.4 (65.5–96.2)

N: weighted study population; %: weighted prevalence; CI: confidence intervals. MMAS = Morisky Medication Adherence Score. Missing value on MMAS (36.4%).

**Table 3 healthcare-10-02244-t003:** Prevalence of low-moderate medication adherence in participants with diagnosed hypertension aged 18 years old or more, stratified by sex and selected variables.

	National N = 622,580 % (95% CI)	Women N = 334,108 % (95% CI)	Men N = 288,474 % (95% CI)
**Sociodemographics**			
Sex			
Men	74.4 (67.5–80.4)	NA	NA
Women	81.4 (78.4–84.0)	NA	NA
Education			
Higher education	75.3 (64.8–83.4)	84.4 (76.8–89.8)	67.2 (48.6–81.6)
Secondary education	81.8 (76.3–86.2)	83.1 (78.4–86.9)	80.0 (68.1–88.2)
Primary education	76.0 (70.3–81.0)	79.0 (73.6–83.6)	72.2 (60.8–81.3)
No formal education	77.2 (65.8–85.6)	76.7 (63.6–86.2)	77.8 (55.6–90.7)
**Condition-related**			
Current smoker	92.0 (75.8–97.7)	85.0 (56.0–96.2)	98.5 (89.4–99.8)
BMI categories			
Normal (18.5–24.9 kg/m^2^)	76.7 (66.6–84.5)	76.9 (66.3–85.0)	76.5 (60.1–87.6)
Overweight (25–29.9 kg/m^2^)	77.1 (71.3–82.1)	81.0 (75.5–85.6)	72.9 (62.2–81.5)
Obese (≥30.0 kg/m^2^)	82.3 (77.0–86.7)	84.6 (80.2–88.2)	78.2 (64.3–87.6)
Alcohol consumption			
Non-drinker	77.6 (73.9–80.9)	81.1 (78.0–83.9)	72.6 (64.7–79.3)
Moderate drinker	86.8 (69.8–94.9)	88.2 (56.6–97.7)	85.9 (61.7–95.8)
Excessive drinker	79.7 (62.6–90.2)	81.1 (64.5–91.0)	79.1 (54.6–92.2)
**Patient-related**			
CKD, CVD, and/or AMI	61.8 (40.0–79.7)	88.7 (70.8–96.2)	52.0 (28.0–75.2)
Anxiety and/or depression	88.6 (73.4–95.6)	89.7 (73.7–96.4)	84.2 (26.8–98.7)
**Time from hypertension diagnosis**			
Antihypertensive medication intake at least two weeks before the interview	86.2 (80.4–90.5)	83.7 (76.7–88.8)	89.2 (78.3–94.9)
Less than a year before the interview	74.4 (62.5–83.6)	79.4 (68.2–87.4)	69.6 (48.4–84.8)
1–5 years before the interview	73.4 (64.9–80.5)	79.1 (72.6–84.4)	66.4 (50.1–79.5)
≥6 years before the interview	77.3 (71.8–81.9)	81.6 (77.1–85.5)	71.5 (60.5–80.5)
**Health checkups frequency**			
None	80.9 (68.0–89.4)	84.8 (73.9–91.7)	77.2 (53.7–90.8)
1–2 yearly	79.6 (75.3–83.3)	80.6 (76.4–84.3)	78.4 (70.1–84.9)
≥3 yearly	74.9 (68.0–80.8)	81.6 (76.6–85.7)	65.9 (52.1–77.4)

High adherence: zero points in the Morisky Medication Adherence Scale. Low-moderate adherence: at least a point in the Morisky Medication Adherence Scale. NA: Not available. N: weighted study population. % = percentage. CI = confidence intervals. BMI = body mass index. CKD: chronic kidney disease. CVD = cerebrovascular disease. AMI = acute myocardial infarction.

**Table 4 healthcare-10-02244-t004:** Crude and adjusted logistic regression models for factors associated with low-moderate medication adherence in hypertensive participants aged 18 years old or more stratified by sex. ENSPA 2019.

	Women		Men	
Sociodemographic	Crude OR (95% CI)	Adjusted * OR (95% CI)	Crude OR (95% CI)	Adjusted * OR (95% CI)
Age in years	0.99 (0.98–1.00)	0.99 (0.98–1.00)	0.99 (0.96–1.01)	1.00 (0.97–1.03)
Regions				
Urban	1 (reference)	1 (reference)	1 (reference)	1 (reference)
Rural	0.67 (0.47–0.97)	0.66 (0.42–1.03)	1.80 (0.97–3.36)	1.77 (0.78–3.98)
Indigenous	1.66 (0.35–8.01)	5.15 (1.40–18.98)	2.55 (0.45–14.51)	2.49 (0.22–27.91)
Education				
Higher education	1 (reference)	1 (reference)	1 (reference)	1 (reference)
Secondary education	0.91 (0.51–1.62)	0.90 (0.48–1.70)	1.95 (0.73–5.23)	2.94 (1.03–8.36)
Primary education	0.70 (0.39–1.24)	0.76 (0.38–1.56)	1.27 (0.50–3.18)	1.43 (0.49–4.13)
No education	0.61 (0.28–1.36)	0.77 (0.28–2.14)	1.71 (0.48–6.06)	0.97 (0.19–5.05)
**Condition-related**				
Current smoker	1.31 (0.30–5.63)	1.81 (0.33–9.85)	24.24 (3.11–118.72)	16.74 (1.83–152.70)
BMI categories				
Normal (18.5–24.9 kg/m^2^)	1 (reference)	1 (reference)	1 (reference)	1 (reference)
Overweight (25–29.9 kg/m^2^)	1.28 (0.69–2.38)	1.35 (0.73–2.50)	0.83 (0.33–2.05)	0.75 (0.32–1.78)
Obese (≥30.0 kg/m^2^)	1.65 (0.90–3.03)	1.49 (0.80–2.77)	1.10 (0.39–3.06)	1.52 (0.59–3.93)
Alcohol consumption				
Non-drinker	1 (reference)	1 (reference)	1 (reference)	1 (reference)
Moderate drinker	1.75 (0.31–9.74)	1.11 (0.20–6.06)	2.29 (0.60–8.79)	2.23 (0.44–11.43)
Excessive drinker	1.00 (0.42–2.40)	0.56 (0.23–1.39)	1.42 (0.43–4.66)	0.93 (0.29–2.97)
**Patient-related**				
CKD, CVD and/or AMI	1.82 (0.57–5.86)	1.73 (0.41–7.41)	0.33 (0.11–0.95)	0.44 (0.15–1.34)
Anxiety and/or depression	2.04 (0.66–6.30)	4.89 (1.36–17.49)	0.78 (0.08–7.85)	0.78 (0.08–7.85)
**Time from hypertension diagnosis**				
Antihypertensive medication intake at least two weeks before the interview	1.15 (0.68–1.94)	1.34 (0.61–2.94)	3.27 (1.26–8.49)	4.35 (1.11–17.11)
Less than a year before the interview	0.87 (0.45–1.65)	1.33 (0.66–2.68)	0.91 (0.33–2.50)	1.14 (0.37–3.52)
1–5 years before the interview	0.85 (0.54–1.34)	1.06 (0.50–2.23)	0.79 (0.34–1.81)	1.12 (0.35–3.62)
≥6 years before the interview	1 (reference)	1 (reference)	1 (reference)	1 (reference)
**Health checkups frequency**				
None	1.26 (0.60–2.65)	1.50 (0.68–3.32)	1.75 (0.52–5.83)	2.97 (0.63–13.88)
1–2 yearly	0.82 (0.53–1.28)	0.82 (0.53–1.28)	1.88 (0.91–3.86)	1.61 (0.78–3.32)
≥3 yearly	1 (reference)	1 (reference)	1 (reference)	1 (reference)

* Logistic regression models were adjusted by each of the other variables. Low-moderate adherence: at least a point in the 4 items Morisky Medication Adherence Scale. OR: odds ratio. CI = confidence intervals. BMI = body mass index. CKD: chronic kidney disease. CVD = cerebrovascular disease. AMI = acute myocardial infarction.

## Data Availability

The datasets used and/or analyzed during the current study are available on reasonable request via email (cnino@gorgas.gob.pa).

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
