# Peer review of "Medication Adherence in Hypertensive Individuals in Panama 2019: A National Cross-Sectional Study"

_healthcare, 2022, doi:10.3390/healthcare10112244_

Round 1

Reviewer 1 Report

Congratulation for the amount of the work you have done. My suggestions are minor comments:

Introduction: line 32: please modify (4) to [4]

Kind suggestion to enrich discussion section.

Reviewer 2 Report

Dear authors, 

congratulations for your work. You tried to acess and discuss the adherence to therapy in Hypertensive individuals. Your manuscript needs to be significantly improved before of being accepted for publication, specialy your discussion. You need to compare your results with several other studies (eg. https://www.ncbi.nlm.nih.gov/pmc/articles/PMC6635171/; https://www.ahajournals.org/doi/10.1161/CIRCRESAHA.118.313220).

Here are my specific suggestions:

Abstract

Add sample data;

Add most important results

Add conclusion

Introduction

LINE 26 you can not state that Hypertension leads to kidney and cardiovascular disease. You can state that is a major risk factor.

Methods:

Line 114 – Add reference to support

Discussion

Line 236-238 – Refer the country where was the study made.

LINE – 242 – 250 – add studies to compare your results

Your discussion needs work. You must discuss your work with similar studies and see similarities and differences between. There are several studies about this thematic.

Conclusion

What is your conclusions?

Reviewer 3 Report

The Authors present a paper: " Medication adherence in Hypertensive individuals in Panama 2019: a national cross-sectional study" worthy for the large number of collected participants. They report the national prevalence of medication adherence in hypertensive individuals  and identification associated factors using data from the National Health Survey for Panama. I consider this as example for other realities.

Introduction, Study Materials, Statistical Analysisas well as the Results are appropriately reported. Now my comments:

1. Outcome assessment: The adherence to therapy assessed through the Spanish version of the semistructured questionnaire could be a bias? please give me a comment on the administartion of the questionnaire . direct interview? telephone call? written response?

2. Discussion is too short compared to the full text. I appreciated the description of the limitations of the study BUT what about the suggestion or recommendation to face the problem? Please give us some also personal remark. It is important not to scare the patient but to empower him. In which way? please don't raise to another study!

3. some tables should be removed as appendix : description in the text is fine.

Round 2

Reviewer 2 Report

Dear authors,

congratulations for your work. You have attended my suggestions and the article is suitable for publlication.